# Evaluation of umbilical cord blood serum eye drops for severe dry eye in two distinct populations—Sjögren's syndrome and mustard gas–induced ocular injury: Protocol for a pilot randomized clinical trial

Seyed Hashem Daryabari[1], Hossein Aghamollaei[2], Seyyed Morteza Hosseini Imeni[3], Masoud Rostami [4]*

1 Trauma Research Center, Clinical Sciences Institute, Baqiyatallah University of Medical Sciences, Tehran, Iran, 2 Chemical Injuries Research Center, Systems Biology and Poisonings Institute, Baqiyatallah University of Medical Sciences, Tehran, Iran, 3 Dida Eye Research Center, Bushehr University of Medical Sciences, Bushehr, Iran, 4 Student Research Committee, Baqiyatallah University of Medical Sciences, Tehran, Iran

* rostamimasoud7595@gmail.com

## Abstract

### Background

Human umbilical cord blood (HUCB) serum eye drops contain growth factors, neurotrophic agents, and antimicrobial compounds that may promote ocular surface healing and regeneration. Sjögren's syndrome is a chronic autoimmune condition characterized by reduced tear production and ocular surface damage, often resulting in severe dry eye symptoms. Mustard gas chemical veterans also suffer from similar debilitating ocular complications due to chronic inflammation and meibomian gland dysfunction. While conventional treatments offer symptomatic relief, they lack essential components of natural tears.

### Materials and methods

This pilot randomized controlled trial will evaluate the efficacy of HUCB eye drops in three patient groups: (A) Sjögren's patients treated with HUCB drops, (B) Sjögren's patients receiving conventional treatment, and (C) mustard gas veterans receiving conventional treatment in the right eye and HUCB drops in the left eye. Patients will be assessed using subjective and objective tools, including the Ocular Surface Disease Index (OSDI), visual acuity, tear film breakup time (TBUT), SM tube test, and fluorescein staining based on SICCA criteria. Group allocation will follow a blocked randomization sequence for groups A and B; group C will follow a within-subject paired-eye design. Follow-up will occur at 30 and 60 days.

**Data availability statement:** Deidentified research data will be made publicly available when the study is completed and published.

**Funding:** The author(s) received no specific funding for this work.

**Competing interests:** The authors have declared that no competing interests exist.

## Discussion

This study aims to assess the regenerative potential of HUCB serum in patients with autoimmune and chemically induced dry eye. Its results may support broader clinical use of HUCB drops in treating severe ocular surface disorders, including conditions like Stevens-Johnson syndrome and industrial chemical exposures.

## Trial registration

This trial was registered at the Iranian Registry of Clinical Trials (Registration number: **IRCT20230925059512N1**, Registration date: **2024-08-11**).

## Background

Sjögren's syndrome is a chronic autoimmune disease that primarily affects exocrine glands, particularly the salivary and lacrimal glands. This leads to dry mouth and eyes as a common symptom in patients. However, it is often associated with musculoskeletal disorders and Inflammatory processes damage to other body systems [1]. The diagnosis of Sjogren's syndrome is based on a combination of dryness symptoms and autoimmune features, including T-cell activity (confirmed by a positive salivary gland biopsy) or B-cell activity (demonstrated by the presence of auto-antibodies) [2]. In most cases, Sjogren's syndrome occurs independently of other inflammatory and autoimmune diseases and is referred to as primary Sjogren's syndrome. However, it can also occur together with other autoimmune diseases such as thyroid disease, rheumatoid vasculitis and systemic lupus erythematosus, which is referred to as secondary Sjogren's syndrome [3]. An increased prevalence of dry eye is observed in patients with autoimmune diseases, including Sjogren's syndrome. Approximately 8% of the population is affected by this disease, 78% of whom are women [4,5]. Dry eye is typically due to instability of the tear film, with the thickness of the lipid layer being a key factor in this stability [6]. Chronic inflammation combined with decreased secretion of tear fluid and increased evaporation contribute to its development [7]. Patients with dry eye may complain of foreign body sensation, pain, sensitivity to light, blurred vision and difficulty opening the eyes. [8] Tears contain various growth factors and essential compounds, including epidermal growth factors, vitamin A, neurotrophic factors, albumin, and specific and non-specific antimicrobial substances such as lysozyme, lactoferrin, alpha-lysine and protein-based antimicrobial components. These elements play an important role in maintaining the health of the epithelial surface of the eye [9,10]. Lysozyme destroys bacteria by hydrolyzing the peptidoglycan layer of bacterial cell walls. In addition, lysozyme also has antifungal properties due to its ability to degrade chitin [10,11]. Alpha-lysine disrupts cell membranes by an unclear mechanism, and its concentration in tears is higher than in serum or plasma [12]. Lactoferrin reversibly binds to two iron atoms, limiting bacterial access to iron, which is critical for their growth and metabolism [13]. Specific antimicrobial components in tears include immunoglobulins A, G and M [14].

The tear film provides a smooth optical surface for the refraction of light, nourishes the cornea and plays a crucial role in the eye's defense system. A lack of tear production or a change in tear composition can lead to eye damage and pathological conditions [15].

Long-term ocular complications resulting from exposure to chemical warfare agents include photophobia (73.2%), perceived visual deterioration (72.5%), dry eye sensation (66.4%), foreign body sensation (61.1%), excessive tearing (46.3%) and pain (43%). Slit lamp findings indicate meibomian gland dysfunction (96%), eyelid inflammation, tear abnormalities and dry ocular surfaces (80–90%). Chronic eyelid inflammation and decreased tear secretion are two key factors contributing to the progression of these ocular complications. The likelihood of chemical-induced keratopathy increases with higher initial exposure, higher degree of disability and longer duration of ocular involvement [16].

Initial ocular symptoms of chemical agent exposure include lacrimation, edema, ocular discharge, and even blindness, which may gradually resolve over the course of days or weeks. However, residual keratopathy may remain, leading to long-term effects in affected individuals (DMGK). Patients often suffer from dry eye symptoms [17].

Current treatments for dry eye syndrome include artificial tears, topical corticosteroids, cyclosporine A, therapeutic contact lenses, protective eyewear, punctal occlusion and tarsorrhaphy. However, these methods do not provide the essential growth factors and components found in natural tears [18].

Eye drops derived from the umbilical cord blood of healthy mothers contain high levels of cytokines, vitamin A, growth factors, neurotrophic factors and essential tear components, as well as antibacterial and antifungal agents such as IgG, lysozyme and other compounds. These drops create a suitable environment for the growth and regeneration of damaged cells and tissues. They are preservative-free, safe and effective in promoting the repair and regeneration of damaged epithelial cells and improving epithelial diseases [9,19,20].

Human umbilical cord blood serum (HUCB) eye drops are a safe and effective treatment rich in growth factors, essential tear components, neurotrophic and defensive factors. These drops not only provide the necessary tear components, but also provide an ideal environment for the regeneration and repair of damaged tissue. With a structural composition very similar to natural tears, they have shown significant efficacy in the treatment of dry eyes and ocular surface disorders.

However, its specific therapeutic effects in patients with Sjogren's syndrome — a condition characterized by combined lacrimal gland dysfunction and ocular surface damage — and in chemical warfare veterans suffering from severe dry eye and extensive ocular surface damage, including corneal and conjunctival lesions, have not yet been thoroughly investigated. Although the underlying causes differ (autoimmune vs. chemical injury), both conditions share common pathogenic mechanisms — including epithelial damage, tear film instability, and chronic inflammation. Therefore, investigating HUCB serum in these two distinct but mechanistically related populations provides an opportunity to evaluate its regenerative potential across different etiologies of severe dry eye.

This study aims to investigate the effects of HUCB eye drops:

1. The improvement and repair of damaged ocular surface tissue.

2. The restoration of vision diminished by damage to the ocular surface tissue.

3. The relief of severe dry eye symptoms in patients with Sjogren's syndrome and chemical warfare veterans.

Both subjective and objective tests will be used to evaluate these outcomes. The results of this study may also have application to other eye conditions associated with severe dry eye and ocular surface damage, such as Stevens-Johnson syndrome, keratoconjunctivitis sicca, burns, and chemical warfare or industrial exposure injuries.

## Materials and methods

### Study design and setting

This randomized, controlled, and open-label clinical trial will be conducted in the Baqiyatallah University of Medical Sciences, Tehran, Iran. Patients will enter the study based on the eligibility criteria and will be followed until 60 days

post-intervention. We began participant recruitment on April 1, 2025, and estimate that the participant recruitment will be completed by July 31, 2025, data collection will end by September 30, 2025, and the results are expected to be ready by October 31, 2025.

This pilot trial includes two distinct yet related populations: (1) patients with Sjögren's syndrome, and (2) mustard gas–exposed veterans with chronic dry eye. The analyses will be conducted independently for these groups, as they represent separate exploratory sub-studies designed to assess feasibility and estimate effect sizes in autoimmune and chemically induced dry eye, respectively.

## Ethical approval

This protocol was approved by the Research Ethics Committee of the Baqiyatallah University of Medical Sciences and is in accordance with the ethical principles and the national norms and standards for conducting medical research in Iran (Approval ID: IR.BMSU.BAQ.REC.1403.080, Approval Date: 2024-06-29). Also, this trial was registered at the Iranian Registry of Clinical Trials (Registration number: **IRCT20230925059512N1**, Registration date: **2024-08-11**).

## Outcome measures

**Primary outcome (dry eye symptoms).** To assess subjective symptoms of dry eye, the Persian version of the OSDI questionnaire, which has been validated and shown to have good reliability and validity in the Iranian population, will be used [21].

**Secondary outcomes. Visual acuity:** The corrected visual acuity of patients will be recorded at each follow-up based on the logarithm of the minimum angle of resolution (log MAR).

**Tear film breakup time test:** The stability of the tear film on the cornea will be assessed using a slit lamp with a cobalt blue filter and sodium fluorescein. A drop of 2% sodium fluorescein will be instilled into the eye, and the patient will be asked to blink five times to spread and form the tear film over the cornea and conjunctiva. The patient is then instructed to refrain from blinking, during which black spots or streaks appear, indicating dry spots and areas where the tear film stability is disrupted. The interval between the last blink and the appearance of the first dry spot will be randomly considered the tear breakup time (TBUT). The average of three measurements will be recorded. A TBUT of less than 10 seconds will be considered abnormal [22].

**SM tube test:** The measurements of the SM Tube may provide a novel, swift, noninvasive, and convenient approach to screen and diagnose DED with acceptable repeatability and reproducibility and specific correlations with TMH, BUT, and SIT [23].

**Fluorescein staining of the cornea and conjunctiva:** According to the SICCA (Sjogren's International Collaborative Clinical Alliance) criteria, the cornea and the nasal and temporal conjunctiva of each eye are considered separate regions. Scoring is based on the extent of staining: the cornea receives a score between 0–6, and each conjunctival region (nasal and temporal) receives a score between 0–3, resulting in a total score range of 0–12 per eye.

For corneal staining:

• A score of 0 is assigned when there is no staining.

• A score of 1 is given for 1–5 staining points.

• A score of 2 is assigned for 6–30 staining points.

• A score of 3 is given for more than 30 staining points.

Additionally, one extra point is added for observing any of the following: a cluster of continuous staining points, staining in the central corneal area, or one or more stained filaments. The total corneal staining score ranges from 0 to 6.

For conjunctival staining (nasal and temporal):

• A score of 0 is assigned for 0–9 staining points.

• A score of 1 is given for 10–32 staining points.

• A score of 2 is assigned for 33–100 staining points.

• A score of 3 is given for more than 100 staining points.

The combined score for nasal and temporal conjunctival staining ranges from 0 to 6 [24].

## Procedure

In brief, all the assessments and evaluations are presented in **Table 1**.

**Recruitment and baseline assessment.** Individuals who visit Baqiyatallah Hospital and meet the criteria for participation in the study will be assessed by a rheumatologist. Those with a confirmed diagnosis of Sjögren's syndrome, based on rheumatological factors and clinical findings, including a combination of dry eye symptoms and autoimmune characteristics (such as T-cell activity, confirmed by a positive salivary gland biopsy, or B-cell activity, confirmed by the presence of autoantibodies), will be referred to the ophthalmology clinic. If they agree to participate, they will be included in the study.

Additionally, individuals with moderate to severe mustard gas chemical eye injuries who exhibit dry eye symptoms, as diagnosed by an experienced ophthalmologist, will also be included in the study.

For the control group, standard treatments for dry eye, such as artificial tears and topical corticosteroid drops, will be used, and the changes will be monitored. A convenience sampling method will be used, and the study's purpose will be clearly explained to participants. Written informed consent will be obtained from all participants.

**Table 1. Project assessment and time steps.**

| Procedures | Screening Day −1 | Intervention Day −1 | Study Visit 1 Day 30 | Study Visit 2 Day 60 |
|---|---|---|---|---|
| Written informed consent | X | | | |
| Demographics | X | | | |
| Medical history | X | | | |
| Randomization | X | | | |
| Administer study intervention | | X | | |
| Concomitant medication review | X | X | X | X |
| Performance status | X | | | |
| REFRACTION (OU) | X | | X | X |
| BCVA (Log MAR) | X | | X | X |
| OSDI questionnaire | X | | X | X |
| TBUT (s) | X | | X | X |
| Tear meniscus height (TMH) | X | | X | X |
| SM tube test | X | | X | X |
| Corneal staining (SICCA): | X | | X | X |
| Conjunctiva staining TEMPO (SICCA) | X | | X | X |
| Conjunctiva staining NASAL (SICCA) | X | | X | X |
| Adverse event review and evaluation | X | X | X | X |
| Complete Case Report Forms (CRFs) | X | X | X | X |

**Umbilical cord serum droplets.** After childbirth, umbilical cord blood was collected directly as part of this study by the research team at Baqiyatallah University of Medical Sciences, Tehran, Iran. The collection took place in affiliated hospitals under ethical approval and informed written consent from the donor mothers. All cord blood samples were obtained in sterile containers without anticoagulants and were immediately processed under controlled laboratory conditions.

We included women aged 18–45 without any current known infection who underwent elective cesarean section. Also, in laboratory conditions, the donated umbilical cord blood samples are screened for contagious viral pathogens such as Human Immunodeficiency Virus (HIV), syphilis, and hepatitis B and C viruses. These samples will be taken in the same timeline as the rest of the study. We began recruitment of eligible donor mothers on April 1, 2025, and estimate that it will be completed by July 31, 2025.

The virus-free umbilical cord blood is incubated for two to four hours at room temperature, followed by the blood centrifugation process. After centrifugation, the yellow supernatant (serum) is carefully separated into a new sterile tube, and the remaining sediment (red blood cells and other cellular debris) should be discarded. Thermal inactivation should then be carried out at 56°C for 30 minutes on the collected serum. Following inactivation, filtration is performed, and the serum is stored at −20°C for future clinical use.

Serum eye drops, including autologous serum eye drops (ASEDs) and umbilical cord blood serum (UCBS), are advanced biologic treatments used to manage severe ocular surface diseases.

Opened serum drops should be stored in the refrigerator at 4°C within 7 days to ensure safety and effectiveness due to the lack of preservatives and risk of contamination, while unopened drops should be kept at −20°C for up to 3 months. The serum drops are administered 6–10 times a day, and if necessary, preservative-free artificial tears may also be recommended [25].

### Eligibility criteria

Patients with Sjögren's syndrome and victims of chemical eye injuries who suffer from severe dry eye symptoms and do not respond well to conventional dry eye treatments will be included in the study. These individuals will have a short tear break-up time (less than 5 seconds), low SM tube results (less than 5 mm), and positive fluorescein staining of the cornea and conjunctiva (≥3).

### Intervention

The participants are divided into three groups:

**Intervention group A:** Patients with definite diagnosis of Sjogren's and dry eyes.
**Control group B:** For the control group, conventional dry eye treatment methods (standard of care).
**Intervention group C:** Mustard gas chemical veterans with dry eye symptoms.

In the group A patients, after performing the desired tests, we will give the patient eye drops made from umbilical cord blood serum and will ask them to put one drop in each eye 6–10 times a day. For the group B, conventional dry eye treatment methods such as artificial tear drops and topical corticosteroid drops will be used and the changes will be monitored. In the group C, patient's right eye will be treated with conventional methods such as artificial tears and Flucort drops, and one drop of cord blood serum will be dropped in the left eye, 6–10 times a day.

### Randomization

A randomized list with block size of 2 will be generated using the Sealed Envelope website [26]. The sequence of participants in the list will be available only to an external observer responsible for participant allocation, that will not participate in the other parts of the study. The randomization will be performed only for groups A and B. Patients in group C will get conventional treatment in their right eyes and umbilical cord serum in their left eye.

## Blinding and study termination

This study will be an open label study that only the assessing optometrist and ophthalmologist will not be aware of the participant allocation and the participants, external observer responsible for participant allocation, and the physician performing the intervention will be aware of the patient allocation.

## Concealment

We will be using sealed envelopes for concealment.

## Sample size

Given the pilot nature of the study, no formal sample size calculation was conducted. However, the results will be used to estimate effect sizes and variability to inform future definitive trials. we will include 10 patients in each group of the study. This number was also determined based on available resources, expected recruitment rates, and the exploratory aim of generating effect size estimates for planning a future definitive trial. Since the number of mustard gas veterans are limited, we will be using each patient of this group as their own controls.

## Statistical analysis

All statistical analyses will be performed using Stata version 17.0. The analysis will focus on estimation rather than formal hypothesis testing, in line with the pilot nature of this trial. The primary goal is to generate effect size estimates and their corresponding 95% confidence intervals to inform the design of a future definitive trial. A p-value of $< 0.05$ will not be interpreted as definitive evidence of a treatment effect due to the small sample size.

Descriptive statistics will be used to summarize baseline characteristics of participants across the three groups. Continuous variables will be reported as mean ± standard deviation (SD), and categorical variables will be expressed as frequencies and percentages.

To estimate the effect of the intervention in group A vs. B comparison, regression-based modeling will be employed. For continuous outcomes, a linear regression model will be used. The β coefficient will represent the adjusted mean difference in the outcome between the HUCB and conventional treatment groups. Its 95% confidence interval will provide a range of plausible effect sizes. For binary outcomes, a logistic regression model will be used. The exponentiated coefficient for the treatment group will be reported as an Odds Ratio (OR) with its 95% confidence interval.

For Group C, which uses a paired-eye design, a linear mixed-effects model will be used to account for the non-independence of two eyes from the same patient. The model will include treatment (HUCB vs. conventional) as a fixed effect and a random intercept for each patient. The primary output will be the estimated mean difference between treatments and its 95% confidence interval.

The effect size and standard deviation of the OSDI score (primary outcome) will be used to inform future sample size and power calculations for a definitive randomized controlled trial. The 95% confidence intervals obtained in this pilot will provide the parameters required to estimate the sample size needed to detect a clinically meaningful difference in OSDI between treatment groups with adequate power.

All analyses will follow the intention-to-treat principle. Efforts will be made to minimize missing data by maintaining regular contact with participants and performing reminder calls before each visit. In case of missing outcome data, multiple imputation using chained equations will be applied if more than 10% of data are missing for the primary outcome (OSDI score). The imputation model will include treatment group, baseline OSDI score, and other relevant covariates.

## Discussion

This randomized controlled trial aims to evaluate the efficacy of human umbilical cord blood (HUCB) serum eye drops in improving ocular surface health, restoring vision, and alleviating severe dry eye symptoms in patients with Sjogren's

syndrome and chemical warfare veterans. By incorporating both subjective and objective outcome measures, this study will provide a comprehensive assessment of HUCB eye drops' therapeutic potential.

The novelty of this study lies in its investigation of HUCB eye drops as a regenerative treatment for severe dry eye, particularly in patients with underlying autoimmune and chemically induced ocular surface damage. Unlike conventional treatments, HUCB serum eye drops offer essential growth factors, neurotrophic components, and antimicrobial agents that closely mimic natural tears, potentially promoting ocular surface repair and regeneration more effectively.

One of the strengths of this study is its randomized and controlled design, which includes validated assessment tools, a well-defined patient population, and a structured follow-up protocol. The inclusion of both Sjogren's syndrome patients and chemical warfare veterans broadens the applicability of the findings, with potential implications for other severe ocular surface disorders, such as Stevens-Johnson syndrome and industrial chemical exposure injuries.

This study is open-label because the physical properties and storage conditions of umbilical cord blood serum differ visibly from conventional artificial tears, making participant blinding infeasible. However, the ophthalmologist and optometrist responsible for clinical assessments will remain blinded to treatment allocation to minimize measurement bias. Furthermore, objective ophthalmic parameters (TBUT, fluorescein staining, SM tube test) are included as secondary outcomes to complement the subjective OSDI score. The absence of participant blinding is acknowledged as a limitation, and results based on subjective measures will be interpreted with caution. Although a longer follow-up period would provide valuable information about the durability of treatment effects, this pilot study was designed primarily to assess short-term feasibility and preliminary efficacy. Due to resource and logistical constraints, extending follow-up beyond 60 days was not possible. The results of this trial will, however, inform the design and timing of follow-up assessments in a future larger-scale RCT with a longer observation period.

In conclusion, this study has the potential to advance treatment options for patients with severe dry eye and ocular surface damage. If successful, the findings may support the integration of HUCB serum eye drops into clinical practice as a safe and effective alternative for managing complex ocular surface diseases.

## Supporting information

**S1 File. SPIRIT checklist.**
(DOC)

**S2 File. Farsi Protocol.**
(DOC)

**S3 File. English protocol.**
(DOCX)

**S4 File. SPIRIT File.**
(DOCX)

## Acknowledgments

During the preparation of this work the authors used ChatGPT (developed by OpenAI, 2025) in order to improve the readability and language of this paper. After using this tool, the authors reviewed and edited the content as needed and take full responsibility for the content of the published article.

## Author contributions

**Conceptualization:** Seyed Hashem Daryabari.

**Data curation:** Hossein Aghamollaei, Masoud Rostami.

**Formal analysis:** Hossein Aghamollaei.

**Investigation:** Hossein Aghamollaei.

**Methodology:** Seyed Hashem Daryabari.

**Project administration:** Masoud Rostami.

**Resources:** Seyyed Morteza Hosseini Imeni.

**Software:** Masoud Rostami.

**Supervision:** Seyed Hashem Daryabari.

**Validation:** Seyyed Morteza Hosseini Imeni.

**Visualization:** Seyyed Morteza Hosseini Imeni.

**Writing – original draft:** Seyed Hashem Daryabari, Hossein Aghamollaei, Seyyed Morteza Hosseini Imeni, Masoud Rostami.

**Writing – review & editing:** Seyed Hashem Daryabari, Hossein Aghamollaei, Seyyed Morteza Hosseini Imeni, Masoud Rostami.

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
