## [Decision Letter · Decision Letter 0]

2 Oct 2025

PONE-D-25-27081
Evaluation effect of umbilical cord blood serum drops in the treatment of severe dry eye in patients with Sjogren's syndrome and mustard gas chemical veterans: Protocol for a randomized clinical trial
PLOS ONE

Dear Dr. rostami,

Thank you for submitting your manuscript to PLOS ONE. After careful consideration, we feel that it has merit but does not fully meet PLOS ONE’s publication criteria as it currently stands. Therefore, we invite you to submit a revised version of the manuscript that addresses the points raised during the review process.

We look forward to receiving your revised manuscript.

Kind regards,

Yu-Chi Liu, MD, PhD

Academic Editor

PLOS ONE

3. We note you have included a table to which you do not refer in the text of your manuscript. Please ensure that you refer to Table 1 in your text; if accepted, production will need this reference to link the reader to the Table.

Additional Editor Comments (if provided):

Reviewers' comments:

Reviewer's Responses to Questions

**Comments to the Author**

1. Does the manuscript provide a valid rationale for the proposed study, with clearly identified and justified research questions?

Reviewer #1: Yes

Reviewer #2: Partly

2. Is the protocol technically sound and planned in a manner that will lead to a meaningful outcome and allow testing the stated hypotheses?

Reviewer #1: Partly

Reviewer #2: Yes

3. Is the methodology feasible and described in sufficient detail to allow the work to be replicable?

Reviewer #1: Yes

Reviewer #2: Yes

4. Have the authors described where all data underlying the findings will be made available when the study is complete?

Reviewer #1: No

Reviewer #2: No

5. Is the manuscript presented in an intelligible fashion and written in standard English?

Reviewer #1: Yes

Reviewer #2: Yes

6. Review Comments to the Author

You may also provide optional suggestions and comments to authors that they might find helpful in planning their study.

Reviewer #1: As the statistical reviewer I will focus on methods and reporting. Overall the approaches are appropriate and clearly reported.

Major

1) I would recommend regression based modeling to estimate ORs and SEs for the outcomes of interest, so they can be used in power calculations (as opposed to p-values)

2) Please follow the CONSORT statement.

3) linked to point 1, please clarify how findings will inform future power calculations and for which outcome.

4) clarify how you will deal with missing data in the trial. multiple imputation?

5) I don't understand why participants are not blinded - a clear justification is needed since this is a major limitation for the primary (self-reported) outcome.

Minor

1) justify the selected sample, is it just down to feasibility? resources?

2) Can you clarify where and how data will be shared?

Reviewer #2: This study protocol addresses a clinically relevant problem, and the use of human umbilical cord blood serum drops represents an innovative approach. This therapy may offer an effective treatment option for severe dry eye in patients with Sjögren’s syndrome and chemical ocular injury by using biologically active components with regenerative potential. However, several methodological and reporting issues need clarification and refinement to strengthen the protocol.

Study design: The trial includes two distinct populations, Sjögren’s patients and mustard gas–exposed veterans. The rationale for including both groups is not clearly explained. Based on the methods, it appears that the analyses for the two populations will be conducted independently. A clearer justification for this hybrid design is needed, and the background section should better support the inclusion of mustard gas–exposed patients.

Title: The current title is somewhat confusing. The study combines a between group RCT (Group A and B) with a within-subject paired-eye design (group C). However, the primary comparison is not directly between the Sjogren's syndrome and mustard gas chemical injury. Consider revising the title to more accurately reflect the study design and objectives.

Sample size: No sample size calculation is provided. The protocol would be strengthened by using preliminary or pilot data to estimate the number of participants needed per group, along with the expected effect size, power, and significance level. Without this information, it is difficult to assess feasibility and statistical robustness.

Follow-up duration: The proposed 60-day follow-up may be too short to capture sustained treatment effects, particularly in chronic conditions such as Sjögren’s syndrome and chemical injury–related dry eye. A longer follow-up period should be considered if feasible.

Outcome measures: The current outcome set is limited to conventional clinical and patient-reported measures. Including additional objective endpoints, such as in vivo confocal microscopy (IVCM), could provide valuable insight into corneal nerve changes and ocular surface regeneration, thereby strengthening the study’s conclusions.

7. PLOS authors have the option to publish the peer review history of their article (what does this mean?). If published, this will include your full peer review and any attached files.

Reviewer #1: No

Reviewer #2: **Yes: **Chang Liu

---

## [Author Response · Author response to Decision Letter 1]

20 Oct 2025

Dear Dr. Yu-Chi Liu,

We sincerely appreciate the time and effort you and the reviewer have dedicated to evaluating our manuscript, "Evaluation effect of umbilical cord blood serum drops in the treatment of severe dry eye in patients with Sjogren's syndrome and mustard gas chemical veterans: Protocol for a randomized clinical trial" (Manuscript ID: PONE-D-25-27081). We are grateful for the constructive feedback, which has helped us improve the quality of our work.

We have carefully addressed all the comments and suggestions provided by the reviewer. Below, we provide a point-by-point response to each remark, detailing the revisions made in the revised manuscript. All changes have been highlighted in the revised version for easy reference in yellow.

Reviewer #1:

We sincerely thank you for your insightful and detailed comments.

Comment 1: "I would recommend regression based modeling to estimate ORs and SEs for the outcomes of interest, so they can be used in power calculations (as opposed to p-values)”

Response 1: We have revised the "Statistical Analysis" section to focus on estimation rather than hypothesis testing. The plan now uses regression modeling to estimate effect sizes (Odds Ratios and mean differences) and their confidence intervals, which will be used to inform the power calculations for a future definitive trial (lines 288-308).

Comment 2: "Please follow the CONSORT statement.”

Response 2: Since this manuscript presents the protocol of an ongoing RCT, the appropriate reporting guideline is the SPIRIT 2013 statement rather than CONSORT, which is intended for completed RCTs with results. The SPIRIT checklist is available as an attachment in the submission portal.”

Comment 3: "Please clarify how findings will inform future power calculations and for which outcome.”

Response 3: As this study is a pilot trial, the main goal is to obtain preliminary estimates of effect size and variability for planning a future definitive RCT. We will specifically use the Ocular Surface Disease Index (OSDI) score, our primary outcome, to calculate the mean difference and standard deviation between groups. These estimates will then be used to perform formal power and sample size calculations for the main trial, targeting a clinically meaningful improvement in OSDI score. We made the necessary amendments (lines 309-313).

Comment 4: "clarify how you will deal with missing data in the trial. multiple imputation?”

Response 4: Given the small sample size and exploratory nature of this pilot trial, we will first aim to minimize missing data through close follow-up and reminders. In the event of missing outcome data, we will conduct the primary analysis using an intention-to-treat (ITT) approach, with multiple imputation by chained equations (MICE) applied if more than 10% of data are missing for the primary outcome (OSDI score). We made the necessary amendments (lines 314-319).

Comment 5: "I don't understand why participants are not blinded - a clear justification is needed since this is a major limitation for the primary (self-reported) outcome.”

Response 5: We fully acknowledge that the absence of participant blinding may introduce potential bias, particularly given that the primary outcome (OSDI) is self-reported. However, blinding participants was not feasible in this pilot study because the appearance, viscosity, and storage requirements of umbilical cord blood serum differ markedly from conventional artificial tears, making it practically impossible to mask the intervention.

To minimize potential bias, we ensured that outcome assessors (ophthalmologist and optometrist) were blinded to treatment allocation, and objective measures (TBUT, fluorescein staining, SM tube test) are also included as secondary outcomes. We will clearly discuss this as a limitation and interpret subjective outcomes with appropriate caution. (lines 337-344).

Comment 6: "justify the selected sample, is it just down to feasibility? resources?”

Response 6: The selected sample size (10 participants per group) was primarily determined based on feasibility and available resources, as this is an exploratory pilot study intended to assess study procedures, recruitment capability, and preliminary effect sizes. Our decision also aligns with methodological recommendations suggesting that pilot RCTs typically include 10–15 participants per group to provide reliable estimates of variability for planning a larger definitive trial.

Therefore, the sample size reflects both practical feasibility and methodological precedent for pilot investigations. We made a clarification about this ( lines 284-286)

Comment 7: "Can you clarify where and how data will be shared?”

Response 7: Thank you for raising this issue. We added a data availibility declaration (lines 362-365).

Reviewer #2

Thank you for taking the time to share your valuable feedback with us.

Comment 1: "The trial includes two distinct populations, Sjögren’s patients and mustard gas–exposed veterans. The rationale for including both groups is not clearly explained. Based on the methods, it appears that the analyses for the two populations will be conducted independently. A clearer justification for this hybrid design is needed, and the background section should better support the inclusion of mustard gas–exposed patients.”

Response 1: By including these two populations, we aim to evaluate whether HUCB serum can promote ocular surface healing in different etiological contexts (autoimmune and chemically induced) thus broadening the potential clinical applicability of this intervention.

Analyses for the two groups will indeed be conducted independently, as they represent two separate sub-studies within the same pilot framework. This hybrid design allows efficient use of study infrastructure while generating preliminary data for both populations to inform future, population-specific definitive trials.

We have revised the Background and Study design sections to clarify this rationale and to better emphasize the relevance of mustard gas–exposed veterans to the study’s regenerative focus (lines 126-130 and 148-152).

Comment 2: "The current title is somewhat confusing. The study combines a between group RCT (Group A and B) with a within-subject paired-eye design (group C). However, the primary comparison is not directly between the Sjogren's syndrome and mustard gas chemical injury. Consider revising the title to more accurately reflect the study design and objectives”

Response 2: We agree that the original title may give the impression of a direct comparison between Sjögren’s syndrome and mustard gas–induced dry eye, which is not the case.

To better reflect the study design and objectives, we have revised the title as follows:

Evaluation of umbilical cord blood serum eye drops for severe dry eye in two distinct populations—Sjögren’s syndrome and mustard gas–induced ocular injury: Protocol for a pilot randomized clinical trial.

In case if you have a title in your mind that better reflects the hybrid nature of our study, we are more than willing to hear it (lines 1-3).

Comment 3: "No sample size calculation is provided. The protocol would be strengthened by using preliminary or pilot data to estimate the number of participants needed per group, along with the expected effect size, power, and significance level. Without this information, it is difficult to assess feasibility and statistical robustness.”

Response 3: As this is a pilot study, its primary aim is to assess feasibility, safety, and variability of the outcomes to inform a future adequately powered randomized controlled trial. For this reason, a formal sample size calculation based on power and significance testing was not performed. Nonetheless, the selected sample size of 10 participants per group was determined according to methodological guidance suggesting that 10–15 participants per arm are adequate in pilot RCTs to estimate variance and feasibility parameters for future trials.

Comment 4: "The proposed 60-day follow-up may be too short to capture sustained treatment effects, particularly in chronic conditions such as Sjögren’s syndrome and chemical injury–related dry eye. A longer follow-up period should be considered if feasible”

Response 4: We fully agree that a longer follow-up period would be ideal to assess the durability of treatment effects, particularly in chronic conditions such as Sjögren’s syndrome and chemical injury–related dry eye.

However, due to resource limitations, logistical constraints, and the exploratory nature of this pilot trial, extending follow-up beyond 60 days is not currently feasible. The 60-day period was selected to allow a practical yet clinically meaningful observation window to evaluate short-term safety, feasibility, and preliminary efficacy signals of the intervention.

We plan to use the findings from this pilot to support a future definitive RCT with a longer follow-up duration to assess sustained outcomes. This clarification has been added to the Discussion section (lines 344-349).

Comment 5: "The current outcome set is limited to conventional clinical and patient-reported measures. Including additional objective endpoints, such as in vivo confocal microscopy (IVCM), could provide valuable insight into corneal nerve changes and ocular surface regeneration, thereby strengthening the study’s conclusions”

Response 5: Although advanced imaging tools such as IVCM could provide valuable results, the present pilot study was designed primarily to assess feasibility and short-term clinical outcomes using established subjective and objective measures. Due to equipment and resource limitations, IVCM was not included in this phase.

Sincerely,

Masoud Rostami

---

## [Decision Letter · Decision Letter 1]

6 Nov 2025

Evaluation of umbilical cord blood serum eye drops for severe dry eye in two distinct populations—Sjögren’s syndrome and mustard gas–induced ocular injury: Protocol for a pilot randomized clinical trial

PONE-D-25-27081R1

Dear Dr. rostami,

We’re pleased to inform you that your manuscript has been judged scientifically suitable for publication and will be formally accepted for publication once it meets all outstanding technical requirements.

Kind regards,

Yu-Chi Liu, MD, PhD

Academic Editor

PLOS ONE

Additional Editor Comments (optional):

Reviewers' comments:

Reviewer's Responses to Questions

**Comments to the Author**

1. Does the manuscript provide a valid rationale for the proposed study, with clearly identified and justified research questions?

Reviewer #1: Yes

Reviewer #2: Yes

2. Is the protocol technically sound and planned in a manner that will lead to a meaningful outcome and allow testing the stated hypotheses?

Reviewer #1: Yes

Reviewer #2: Yes

3. Is the methodology feasible and described in sufficient detail to allow the work to be replicable?

Reviewer #1: Yes

Reviewer #2: Yes

4. Have the authors described where all data underlying the findings will be made available when the study is complete?

Reviewer #1: Yes

Reviewer #2: Yes

5. Is the manuscript presented in an intelligible fashion and written in standard English?

Reviewer #1: Yes

Reviewer #2: Yes

6. Review Comments to the Author

You may also provide optional suggestions and comments to authors that they might find helpful in planning their study.

Reviewer #1: I am satisfied with the authors' responses and the resulting changes to the paper. Nothing else to add.

Reviewer #2: Thank you for the authors’ revisions. The authors have addressed and explained my previous comments, and the manuscript has been improved. As a study protocol, I have no further major concerns and recommend the manuscript for publication after minor editorial review.

7. PLOS authors have the option to publish the peer review history of their article (what does this mean?). If published, this will include your full peer review and any attached files.

Reviewer #1: No

Reviewer #2: No

---

## [Editor Report · Acceptance letter]

PONE-D-25-27081R1

PLOS ONE

Dear Dr. Rostami,

I'm pleased to inform you that your manuscript has been deemed suitable for publication in PLOS ONE. Congratulations! Your manuscript is now being handed over to our production team.

Kind regards,

on behalf of

A/Prof Yu-Chi Liu

Academic Editor

PLOS ONE